# Korean Medicine Clinical Practice Guideline Update for Temporomandibular Disorders: An Evidence-Based Approach

**DOI:** 10.3390/healthcare11162364

**Published:** 2023-08-21

**Authors:** Hyungsuk Kim, Jae Woo Shim, Woo-Chul Shin, Yoon Jae Lee, In-Hyuk Ha, Koh-Woon Kim, Jae-Heung Cho

**Affiliations:** 1Department of Korean Rehabilitation Medicine, College of Korean Medicine, Kyung Hee University, Seoul 02447, Republic of Korea; kim0874@khu.ac.kr (H.K.); saewoo0913@khu.ac.kr (J.W.S.); eddyshin41@khu.ac.kr (W.-C.S.); omdkimkw@khu.ac.kr (K.-W.K.); 2Jaseng Spine and Joint Research Institute, Jaseng Medical Foundation, Seoul 06110, Republic of Korea; goodsmile@jaseng.org (Y.J.L.); hanihata@jaseng.org (I.-H.H.)

**Keywords:** temporomandibular disorders, clinical practice guideline, therapeutic options, conservative therapy, systematic review, meta-analysis, Korean medicine

## Abstract

Many updated clinical research results have been published since the Korean Medicine Clinical Practice Guidelines (KMCPGs) for temporomandibular disorders (TMDs) were published in 2018. Therefore, it is necessary to update the existing clinical practice guidelines (CPGs). This study presents updated recommendations for TMD treatment based on current research data published up to February 2020. The draft version of the level of evidence and grade of recommendation was determined through an assessment of the risk of bias and a meta-analysis of selected literature based on the Grading of Recommendations Assessment, Development, and Evaluation (GRADE). The final guidelines were derived using the Delphi method. Eighteen recommendations were derived for eight items of Korean medicine treatment for TMDs. Compared with previous Korean medicine CPGs for TMDs, the grades of seven recommendations, including acupuncture, pharmacopuncture, and Chuna manual therapy, were increased. The grades of the two recommendations have been changed. Six new recommendations were added to fully reflect clinical reality. Acupuncture, pharmacopuncture, and Chuna manual therapy are recommended for TMD patients in clinical practice. Concurrent conventional conservative therapy with Korean medicine or a combination of Korean medicines should be considered in clinical practice in patients with temporomandibular disorders.

## 1. Introduction

Korean medicine, a Traditional East Asian Medicine, was developed based on clinical experience and numerous studies [1]. It has a unique system of diagnosis and treatment and can be complementary and an alternative to conventional medicine [2].

The Korean Medicine Clinical Practice Guideline (KMCPG) development project was conducted as part of the Korean Medicine Leading Technology Development Project conducted in 2016 by the Ministry of Health and Welfare to standardize and scientifically strengthen Korean medicine. This project developed the KMCPG for temporomandibular disorders (TMDs). Based on the reference search results for Korean medicine treatment for TMDs up to October 2016, the 2017 KMCPG for temporomandibular conditions [3] was developed, and preliminary certification was completed. Subsequently, the 2020 KMCPG for TMDs was revised by reflecting on the reference search for newly published studies and the results of clinical research.

Through an analysis that included a newly published reference, an upgraded grade of recommendation was derived for acupuncture, exercise therapy, and Korean physiotherapy treatment for TMDs. Pharmacopuncture and Chuna manipulation have been widely used in Korean clinical practice for TMDs. However, a clear recommendation could not be drawn up because of the very small number of evidence documents or low level of evidence. Therefore, to develop the KMCPG, a clinical study [4,5] was conducted to create evidence for pharmacopuncture and Chuna manipulation. Based on the study’s results, an upgraded grade of recommendation was derived for pharmacopuncture and Chuna manipulation for the treatment of TMDs.

In clinical practice, many patients receiving the usual conservative treatment visit a Korean medicine clinic or hospital for concurrent treatment. In addition, when patients with TMDs visit a Korean medicine hospital, most of them are treated with a combination treatment of Korean medicine. Reflecting on this clinical practice situation, a new clinical question was developed for concurrent and combination treatments in Korean medicine, and new recommendations were added through a reference search and review.

It has been reported that 5–12% of the population has TMD [6,7,8], and the number of patients visiting medical institutions to treat TMDs in Korea is increasing [9]. According to the Healthcare Big Data Hub of the Health Insurance Review and Assessment Service (http://opendata.hira.or.kr/op/opc/olap4thDsInfo.do (accessed on 1 December 2021)), more than 420,000 patients with TMD in Korea in 2015 and more than 480,000 patients with TMD in 2019 were treated. The total medical expenses for TMDs also increased by 41% from KRW 35 billion (USD 27 million) in 2015 to KRW 49.3 billion (USD 38 million) in 2019.

The treatment of TMDs includes patient education, behavioral therapy, physical therapy, and drug therapy [10,11,12]. According to a survey of Korean medicine doctors [13], acupuncture, Chuna manipulation, pharmacopuncture, oral balance devices, and herbal medicine are the most commonly used treatments for TMDs. Studies of Korean medical treatments for TMDs are also being conducted. Representative studies have shown the effectiveness of Korean medical treatments, such as acupuncture, for TMDs [14,15]. However, when searching for existing guidelines for TMDs at home and abroad, there are no guidelines that apply the guideline development methodology, including the diagnosis and treatment in Korean medicine. Therefore, it was concluded that developing a KMCPG for TMDs using evidence-based medical methods is necessary [16].

The KMCPG uses evidence-based research methodology, such as the Grading of Guideline Assessment, Development, and Evaluation (GRADE) [17], which evaluates papers based on the level of evidence and grade of recommendation. The KMCPG for TMDs was developed to reflect the treatment experiences of Korean clinicians through surveys and interviews. Through the KMCPG, we aim to help Korean medicine doctors standardize treatment and decision making, provide standardized, high-quality Korean medicine treatment to patients with TMDs, and ultimately improve the quality of Korean medicine treatment.

This project is a revision of the KMCPG for TMDs [3], which was pre-certified in 2017. This supplemented the limitations of previous guidelines and added new evidence such as evidence-creating clinical research results. In addition, the KMCPG was revised by adding more appropriate recommendations for clinical practice situations, such as concurrent treatment.

## 2. Materials and Methods

### 2.1. Targets, Users, and Healthcare Environments for KMCPG

The KMCPG for TMDs developed in this study targets patients with various symptoms including temporomandibular joint pain, joint disorders, headaches, and facial asymmetry caused by the temporomandibular joint. The primary users of these guidelines are Korean medical doctors who treat patients with TMDs in primary care settings. They can also be used by residents to learn about Korean medical treatments. Therefore, it can be applied in the outpatient and inpatient care settings of the Korean Medicine Clinic and the Superior General Hospital, where Korean medicine doctors treat patients with TMDs.

### 2.2. Formation of the Development Group

The development committee was comprised of clinical and methodological experts. The development committee carried out the entire process necessary for producing medical guidelines, such as developing clinical questions, searching for and analyzing evidence, deriving recommendations, and preparing and writing the contents of the KMCPG.

The review committee, composed of experts in various fields, conducted a preliminary review of the clinical question development and search stage and externally reviewed whether the clinical status was well reflected in the developed clinical question and searched well methodologically. In addition, the draft-created recommendation was reviewed, and the final recommendation was drawn by participating in a formal consensus using the Delphi technique.

### 2.3. Utilization and Dissemination of Korean Medicine Clinical Practice Guideline

For the effective dissemination of recommendations, we developed a tool for guideline summaries and explanatory materials for patients. The KMCPG, which was finally certified by the KMCPG project group, the developed guideline leaflet, and patient explanatory material can be provided through the portal of the National Clearinghouse for Korean Medicine (http://www.nckm.or.kr) and bulletin boards of the Korean Medical Association.

### 2.4. Renewal Plan for Korean Medicine Clinical Practice Guideline

The KMCPG undergoes renewal every five years by the Evaluation Committee of the Society of Korean Medicine Rehabilitation of the KMCPG. This renewal is prompted by the discovery of new scientific evidence or evaluation of user preferences and clinical variations in the recommended interventions. When the need for renewal is identified, the development committee is reconstituted, with a focus on retaining existing members. Furthermore, the process of updating recommendations involves thorough evidence retrieval and analysis.

### 2.5. Declaration of Interest

KMCPG was created as part of the “KMCPG for TMD adaptation and clinical study (HB16C0059)” project of the Ministry of Health and Welfare’s Korean Medicine Leading Technology Development Project. Development funding agencies did not affect KMCPG content. All members who participated in developing the KMCPG made a declaration of conflict of interest to disclose their actual and economic interests with specific institutions or research-related parties and to minimize their interests. In some cases, the clinical trials of the members of the development committee were included in the analysis. However, in this case, the paper was excluded from the selection, risk of bias, and research related to the relevant clinical questions to prevent bias due to conflicts of interest. 

### 2.6. Selection of Key Questions

The key question of the KMCPG for TMDs was “Does the Korean Medicine treatment method show improvement in pain, function, and quality of life in an adult patient with TMD compared to the control group?” The selection criteria for PICO (Patient or Population, Intervention, Comparison or Comparator, Outcome) constituting the key questions are described below.

Patients diagnosed with TMDs or those with symptoms related to TMDs were selected, and adult patients with TMDs aged 19–70 years were included. 

Acupuncture, laser therapy, pharmacopuncture, herbal medicine, Chuna manipulation, exercise, intraoral balance devices, thread-embedding acupuncture, and Korean medicine physical therapy were selected as the interventions. In a survey of Korean medical doctors [13], the treatment method used for TMDs was mainly reflected in the selection of interventions. For acupuncture-related clinical questions, intervention-related options included acupuncture, warm-needle acupuncture, fire-needle acupuncture, and electroacupuncture. Dry needling was also included as acupuncture-related evidence.

For comparison, an inactive control group, such as sham acupuncture or no treatment, was set according to the key question, or the usual conservative treatment, such as analgesics, hyperthermia, and electrophysiotherapy, was selected.

This study attempted to utilize all Outcome variables related to TMDs such as pain, function, quality of life, and improvement rate.

The study design of the supporting references was based on the clinical practice guidelines (CPGs), systematic review (SR), meta-analysis, and randomized controlled trials (RCTs). Observational studies were also included for patients with insufficient safety-related information.

### 2.7. Reference Search and Selection

Core databases (DBs) such as Ovid-Medline, Ovid-EMBASE, and Cochrane Library were searched for comprehensiveness of the reference search. In addition, Ovid-AMED, an alternative-medicine-related DB, was also searched. Furthermore, we searched the China National Knowledge Infrastructure (CNKI) database and the Japanese CiNii database. To fully include domestic references, five domestic DBs were included: OASIS, NDSL, KISS, KoreaMed, and KMBASE. Studies published until February 2020 were searched. The languages used were English, Chinese, Japanese, and Korean. The search formula consisted of “Patient OR Intervention” as the basic structure and was modified and used according to the DB environment. To increase the sensitivity of the search, comparisons and outcomes were excluded.

Two or more researchers independently conducted reference selection for all the searched documents. After duplicate exclusions, the first selection was excluded after checking the title and abstract, and the secondary selection was excluded after reviewing the original text. In cases of disagreement, a consensus was reached after sufficient discussion. In case of a dispute, a third party intervened to reach an agreement. The PICO, determined in advance for each clinical question, was used as the selection criterion. The exclusion criteria were as follows: (1) instances where Korean interventions were administered in manners not consistent with Korean medicine doctors in Korea; (2) a study comparing the control group, which is difficult to see with usual conservative treatment; (3) studies conducted on children and adolescents; and (4) a simple handwriting study that was not based on Korean medicine theory.

### 2.8. Assessing the Risk of Bias in the Reference

A measurement tool to assess the methodological quality of systematic reviews (AMSTAR) [18] was used for the systematic review and meta-analysis. The risk of bias tool of the Cochrane Collaboration was used for randomized controlled trials (RCTs) [19]. Two or more researchers independently evaluated all the research data. In case of inconsistencies in the evaluation, the two evaluators reached an agreement. In cases of disagreement, a third party intervened to reach an agreement.

### 2.9. Synthesis and Analysis of Evidence

If quantitative analysis was possible, a meta-analysis was performed using Review Manager 5.3 (Copenhagen: The Nordic Cochrane Center, The Cochrane Collaboration, 2014). The study was divided according to the comparison group and outcome variables. If the outcome variable was continuous, it was presented as the standardized mean difference (SMD) or mean difference (MD) using inverse variance analysis. If necessary, a subgroup analysis was performed considering the study design (type of comparison group, treatment period, and frequency). Finally, the heterogeneity of the studies was confirmed using the I2 test and chi-square test. The results of the meta-analysis are presented in the Appendix A.

### 2.10. Level of Evidence and Grade of Recommendation

The level of evidence and grade of recommendation were evaluated according to GRADE [17]. The determination of the level of evidence was set to “High” in the case of randomized controlled trials (RCTs) in the study design and “Low” in the case of observational studies. In randomized controlled trials (RCTs), the level of evidence is determined by evaluating factors such as the risk of bias, inconsistency, non-direction, and imprecision. The recommended grade is determined based on the level of evidence and clinical significance. Furthermore, it is evaluated by comprehensively considering the benefits and harms, clinical utility, value and preference, resources, and economic evidence when treatment is administered to the patient [20].

The level of evidence was determined by dividing it into four grades: “High”, “Moderate”, “Low”, and “Very Low” (Table 1), and the grade of recommendation was determined by dividing it into five parts: A, B, C, D, and Good Practice Point (GPP) (Table 2).

### 2.11. Deriving a Final Agreement

The questionnaire was designed based on the draft recommendations prepared by the development committee. A consensus was reached on the recommendation, grade of recommendation, and level of evidence using the Delphi technique, an official consensus method. The external expert panel consisted of 11 people, including three experts from the Korean Society of Acupuncture and Medicine, three experts from the Society of Korean Medicine Rehabilitation, three clinical Korean medicine doctors, including two practitioners, and two experts in guideline methodologies. In the case of very inappropriate responses, 1 point was indicated, and in the case of very appropriate responses, 9 points were indicated, so the responses were scored on a scale of 1–9. If it was analyzed with a median score of 7 or higher, a consensus was reached by agreeing with the recommendations. In the 1st Delphi, it was confirmed that consensus was reached on all recommendations; therefore, the derivation of the consensus ended with the 1st questionnaire. 

## 3. Results

### 3.1. Acupuncture

#### 3.1.1. Acupuncture Group vs. Inactive Control Group

It is recommended that acupuncture be used in clinical practice to treat patients with TMDs (R1, grade of recommendation: A; level of evidence: high). 

Upon analyzing nine studies included in the meta-analysis [21,22,23,24,25,26,27,28,29], it was observed that the acupuncture group exhibited a statistically significant reduction in pain change as measured by the visual analog scale (VAS) in comparison to the inactive control group, including sham with an SMD of 0.46 (95% CI 0.20, 0.72, *p* = 0.0005). Heterogeneity was not confirmed in the meta-analysis (chi-square = 0.53 and I2 = 0%). Therefore, there was no reason to lower the level of evidence. Consequently, the level of evidence was determined to be high.

The meta-analysis results of four studies [21,25,29,30] showed that the acupuncture group had a significantly increased mouth opening compared to the sham acupuncture group (SMD 0.46 [95% CI 0.09, 0.83]). Heterogeneity was confirmed as chi-square = 0.02, I2 = 70%; therefore, the level of evidence was lowered by one level, and the level of evidence was determined to be moderate.

A meta-analysis was performed with one study reporting an improvement rate [31], and the acupuncture group had a risk ratio (RR) of 7.00 (95% CI 1.91, 25.62), which was more effective compared to the sham acupuncture group. However, as only one study was included, and the number of subjects was small, the level of evidence was determined to be moderate by lowering the aspects of imprecision by one level. The results of the meta-analysis are presented in Appendix A.

In a survey of Korean medicine doctors [13], acupuncture was identified as the most widely used treatment for TMDs in adults at Korean medicine clinics and hospitals. Unfortunately, safety could not be evaluated because adverse events were not reported in many of the included studies. However, considering the results of several studies related to the overall safety of acupuncture [29,30], safety concerns were not high. Therefore, the recommended grade was assigned A.

#### 3.1.2. Distal Acupoints Group vs. Local Acupoints Group vs. Concurrent Treatment of Distal and Local Acupoints Group

When treating patients with TMDs, distal and local blood collection should be considered according to the judgment of the Korean medicine doctor, with reference to the patient’s clinical features (R1-1, grade of recommendation: B; level of evidence: moderate).

One study, comparing the local and distal acupoints, was identified [32]. Patients were divided into three groups for comparative analysis: local acupoints, distal acupoints, and distal and local parallel acupoints. All three groups showed significant pain reduction and functional recovery after acupuncture compared to baseline, and the meta-analysis results showed no significant differences between the three groups (Appendix A). However, only one study was included, and the level of evidence was lowered by one notch because of concerns about imprecision owing to the small number of study subjects. Therefore, the level of evidence was considered to be moderate.

Treatment with distal acupoints, local acupoints, and concurrent treatment with distal and local acupoints showed similar clinical benefits. It was judged that there was no difference in the harm; therefore, the grade of recommendation was assigned as B.

#### 3.1.3. Acupuncture Group vs. Usual Conservative Treatment Group

The use of acupuncture should be considered in clinical practice for patients with TMDs (R2, grade of recommendation: B; level of evidence: moderate).

Sixteen studies were identified [24,33,34,35,36,37,38,39,40,41,42,43,44,45,46,47], and eleven were included in the meta-analysis [24,36,39,40,41,42,43,44,45,46,47]. After analyzing the improvement rate in the meta-analysis, nine studies [36,40,41,42,43,44,45,46,47] were included, and the acupuncture group showed a significant improvement rate (RR 1.16 [95% CI 1.03, 1.31]). Heterogeneity was confirmed by chi-square and I2 tests. Due to concerns about inconsistency, the level of evidence was evaluated as moderate. In meta-analysis results for pain, significant pain reduction was confirmed with an MD of −2.32 (95% −4.51, −0.12). The results of the meta-analysis are presented in Appendix A. Although heterogeneity was not approved, the level of evidence was lowered by one notch, as there were concerns about imprecision due to the small number of subjects in the included clinical studies. Therefore, the level of evidence was considered to be moderate.

As shown in the survey of Korean medicine doctors [13], acupuncture is a representative treatment method in the clinical field for adult patients with TMDs in Korean medicine clinics and hospitals. Considering the improvement rate and pain-reducing effect, acupuncture for TMDs has great clinical benefits [48,49]. According to the results of safety studies of acupuncture, safety concerns are low. Therefore, the recommended grade was assigned B.

#### 3.1.4. The Concurrent Treatment Group of Acupuncture and Usual Conservative Treatment vs. Usual Conservative Treatment Group

In clinical practice, acupuncture should be considered for symptom improvement in patients with TMDs receiving conservative treatment (R3, grade of recommendation: B; level of evidence: moderate).

Five papers were selected [33,34,50,51,52], and three were included in the meta-analysis [50,51,52]. The meta-analysis results of two papers [50,52] reporting the improvement rate were RR 1.26 (95% CI 1.05, 1.49), and the concurrent treatment group of acupuncture and usual conservative treatment showed a significant improvement rate compared to the control group. In addition, in the meta-analysis results for pain, significant pain reduction was confirmed with an MD of −1.23 (95% −1.79, −0.67). Meta-analysis results are presented in Appendix A. However, there were concerns about imprecision in both meta-analyses because the numbers of included studies and study subjects were small. Therefore, the level of evidence was considered to be moderate.

Considering that acupuncture is used in the clinical field and that its safety is more significant than its side effects, the grade of recommendation was determined to be B.

### 3.2. Laser Therapy

#### Laser Therapy Group vs. Inactive Control Group

Laser therapy may improve symptoms during the clinical treatment of patients with TMDs (R4, grade of recommendation: C; level of evidence: low). 

Four clinical studies [53,54,55,56] were identified. In the pain meta-analysis results confirmed by VAS, the laser therapy group showed a significant decrease in MD of −1.31 (95% CI −2.17, −0.45) compared to the inactive control group. Chi-square *p* = 0.0009 and I2 = 86% confirmed the heterogeneity of concern. Meta-analysis results confirmed that the amount of mouth opening showed no significant difference with an MD of 3.50 (95% CI: −2.52, 9.52). The results of the meta-analysis are presented in Appendix A.

In terms of pain, laser therapy showed a significant effect compared with the inactive control group; however, there were concerns about inconsistencies due to heterogeneity. There were also concerns regarding imprecision because of the small number of patients included. Therefore, the level of evidence was lowered by two notches and was determined to be low.

As laser therapy did not have high safety concerns, it was judged that the benefits of laser therapy outweighed the harm. Although its utility in the medical field is not currently high, it can be easily applied to patients who are afraid of acupuncture. In addition, as the number of overseas studies increased, the recommendation grade was assigned as C.

### 3.3. Pharmacopuncture

#### Pharmacopuncture Group vs. Usual Conservative Treatment Group

In clinical practice, pharmacopuncture is recommended to improve symptoms in patients with TMDs (R5, grade of recommendation: A; level of evidence: moderate).

Two studies [5,57] were identified; however, the study conducted in China [57] was often different from the pharmacopuncture procedure performed in Korean clinical practice, such as injecting painkillers or vitamins mixed into acupoints during the pharmacopuncture procedure. Therefore, they were excluded from the study based on the derivation of this recommendation.

In the meta-analysis of pain measured by the VAS, the pharmacopuncture group had an MD of –11.77 (95% CI –21.10, –2.44), and a significant pain reduction effect was confirmed compared to the control group. Meta-analysis for pain NRS also confirmed considerable pain reduction with an MD of –1.17 (95% CI −2.10, −0.24). In the meta-analysis of discomfort measured by the NRS, a significant decrease in discomfort was confirmed with an MD of –1.41 (95% CI: –2.37, –0.45). In a meta-analysis of maximal mouth opening volume and quality of life, there was no significant difference between the pharmacopuncture and control groups. Appendix A presents the results of the meta-analysis.

Only one study with 82 study subjects was included in the meta-analysis, and the level of evidence was determined to be moderate, owing to concerns about imprecision [58]. Based on the results of safety reports on pharmacopuncture, the incidence of side effects due to pharmacopuncture is low, and many are non-serious. Therefore, the clinical benefits of pharmacopuncture in TMDs outweigh its harmful effects. In addition, as a result of the economic evaluation, pharmacopuncture was found to have a low cost and high utility from the healthcare system and social perspectives. Therefore, the level of evidence supporting the use of pharmacopuncture was moderate. Still, the grade of recommendation was determined as “A” by raising the recommendation level in one step, considering the economic evaluation results and clinical utility.

Bee venom pharmacopuncture is actively used in clinical practice depending on the patient’s characteristics and history of TMD [13]. However, as evidence related to bee venom pharmacopuncture is limited, we considered bee venom pharmacopuncture in clinical settings. The treatment site was determined according to the protocol of the selected clinical study and the clinical application method. Through this process, it was derived that acupoints such as Imun (耳門, TE21), Hagwan (下關, ST7), and Yepung (翳關, TE17) can be considered as pharmacopuncture acupoints for TMDs as clinical considerations.

### 3.4. Chuna Manipulation

#### 3.4.1. Chuna Manipulation Group vs. Usual Conservative Treatment Group

Chuna manipulation is a type of manual therapy that originated in Korean medicine and includes manipulation, myofascial release, and joint mobilization. Symptoms should be improved during clinical treatment of patients with TMDs (R6, grade of recommendation: A; level of evidence: high).

Five studies [4,59,60,61,62] were identified. Three studies [4,60,62] were included in the meta-analysis of pain VAS, which confirmed that the Chuna manipulation group had significantly reduced pain (MD –1.17 (95% CI –1.71, –0.64)). No significant heterogeneity was identified in the meta-analysis (chi-square test, 0.15; I2 = 48%). The included studies did not have a high risk of bias; therefore, the level of evidence was determined to be high. In a meta-analysis of the extent of mouth opening, evaluating the change in function and quality of life, Chuna manipulation showed significant improvement compared to the usual conservative treatment. However, owing to inconsistencies or imprecision, the level of evidence was considered moderate. The meta-analysis results are presented in Appendix A.

In a safety study, when an adverse event that was presumed to be relevant to the clinical study was identified, the same number of adverse events were observed in the control and treatment groups, and all were mild [4]. In addition, as a result of economic evaluation, the Chuna manipulation was found to have low cost and high utility from the healthcare system and social points of view. Referring to these results, it was judged that the benefits of Chuna manipulation were more significant than the harms and that it was cost-effective; therefore, the grade of recommendation for Chuna manipulation alone was given as A.

In a clinical study of Chuna manipulation conducted in Korea [4,63,64,65], the most commonly used technique was the seated TMJ distraction technique with the thumb and the JS supine position cervical spine distraction correction technique. The supine position cervical distraction and seated lateral pterygoid distraction techniques with the index finger were used most frequently. Clinical considerations were based on these findings.

#### 3.4.2. The Concurrent Treatment Group of Chuna Manipulation and Usual Conservative Treatment vs. the Usual Conservative Treatment Group

In the clinical treatment of patients with TMDs, concurrent treatment with Chuna manipulation should be considered to improve the symptoms in patients receiving conventional conservative treatment (R7, grade of recommendation: B; level of evidence: moderate). A total of two studies [66,67] were identified and were included in the meta-analysis of the improvement rate. With RR 1.28 (95% CI 1.09, 1.49), the concurrent treatment group with Chuna manipulation showed significant improvement (Appendix A). However, because the number of studies included in the meta-analysis and the number of patients were small, the level of evidence was downgraded to moderate owing to concerns about imprecision.

Based on the general clinical situation, the risk of side effects due to Chuna manipulation was not high; therefore, it was considered that the clinical benefit was greater than the harm. Furthermore, Chuna manipulation is the most representative treatment method for TMDs in clinical practice [13]; therefore, it is actively used in the clinical field. Subsequently, the recommended grade was determined as “B”.

#### 3.4.3. Concurrent Treatment Group of Chuna Manipulation and Korean Medicine Treatment vs. Acupuncture or Herbal Medicine Treatment Group

In the clinical treatment of temporomandibular joint patients undergoing acupuncture, Chuna manipulation should be considered to improve symptoms (R8, grade of recommendation: B; level of evidence: moderate). 

Four studies [68,69,70,71] were included in the meta-analysis of improvement rates. With an RR of 1.21 (95% CI 1.10, 1.32), the Chuna manipulation concurrent treatment group showed a significant improvement rate compared with the Korean medicine treatment group without Chuna manipulation. However, there were concerns about selection bias in the assessment of the risk of bias; therefore, we lowered the level of evidence by one level and decided to make it moderate. One meta-analysis of pain reduction included two clinical studies [71,72]. The degree of pain reduction in the concurrent treatment group with Chuna manipulation was more significant (MD 0.24 [95% CI, 0.12, 0.35]). However, there were concerns about imprecision owing to the small number of studies and patients included. The meta-analysis results are presented in Appendix A.

No adverse event-related reports were found in any of the included studies. However, based on the clinical experience of the development committee, it was judged that the concurrent treatment with Chuna manipulation and Korean medicine, such as acupuncture, did not raise safety concerns. Thus, the clinical benefit was higher than the harm. In addition, a survey of Korean medicine doctors [13] confirmed that it is very active in clinical use. Therefore, the recommendation grade was determined to be “B”.

### 3.5. Herbal Medicine

#### 3.5.1. Herbal Medicine Treatment Group vs. usual Conservative Treatment Group

In clinical practice, the use of herbal medicines for symptomatic improvement in patients with TMDs should be considered (R9, grade of recommendation: B; level of evidence: moderate). 

One document was identified, and in this study [73], a total of 150 patients were divided into herbal medicine treatment groups based on four pattern identifications. Different herbal medicines were administered for each symptom, and the improvement rate was compared with a control group aided with analgesics.

A meta-analysis of the improvement rate confirmed significant improvement in the herbal medicine treatment group with an RR of 1.27 (95% CI 1.04, 1.55) (Appendix A). However, because only one study was included, there were concerns about imprecision; therefore, the level of evidence was lowered by one notch, and accordingly, the level of evidence was determined to be moderate.

The herbal medicine prescriptions used in the Chinese reference used as the source reference [72,73] such as Dokhwaljihwang-tang (獨活地黃湯), Soyo-san-gagambang (逍遙散 加減方) were commonly used herbal medicine prescriptions in Korea, so there were no concerns about non-directivity. In addition, herbal medicines prescribed by Korean medicine doctors do not have many side effects [74]. Minor digestive disorders and minor reversible side effects were investigated. Therefore, it was judged that the clinical benefit was more significant than the harm, and the recommendation grade was determined to be “B”, considering that the clinical utility was also high.

Based on clinical references [73,75], as a clinical consideration, the development committee proposed Ssang-hwa-tang (雙和湯), Gal-geun-tang (葛根湯), Man-geum-tang (萬金湯), and Gamisoyosan (加味逍遙散) as herbal medicines for the treatment of TMDs.

#### 3.5.2. Concurrent Treatment Group of Herbal Medicine Treatment and usual Conservative Treatment vs. Usual Conservative Treatment Group

In the clinical treatment of patients with TMDs, herbal medicines should be considered to improve symptoms in patients receiving conservative treatment (R10, grade of recommendation: B; level of evidence: moderate).

Meta-analysis results on the improvement rate of two studies [76,77] were included, and the concurrent treatment group with herbal medicine showed a significant improvement (RR 1.37 [95% CI: 1.17, 1.60]). In addition, a meta-analysis of pain was also analyzed as an MD of −1.29 (95% CI −1.42, −1.16), confirming a significant pain reduction effect. The results of the meta-analysis are presented in Appendix A. In both meta-analyses, because the number of studies and the number of included subjects were small, there may be concerns about imprecision; therefore, the level of evidence was lowered by one notch, and moderate was decided.

Based on the general clinical situation and considering the safety reports related to herbal medicine, it was believed that the concurrent treatment with usual conservative treatment and herbal medicine did not significantly raise safety concerns [74]. Furthermore, the risk of side effects due to herbal medicine prescriptions by Korean medicine doctors was not high; therefore, it was judged that the clinical benefit outweighed the harm. Accordingly, the recommended grade was determined as “B”.

#### 3.5.3. Concurrent Treatment Group of Herbal Medicine Treatment with Korean Medicine Treatment vs. Korean Medicine Treatment

In the clinical treatment of temporomandibular joint patients undergoing acupuncture, herbal medicines should be considered to improve symptoms (R11, strength of recommendation: B; level of evidence: moderate).

Meta-analysis results on the improvement rate in two studies [78,79] confirmed that the herbal medicine concurrent treatment group showed significant improvement (RR 1.25 [95% CI 1.01, 1.54]). In addition, meta-analysis of pain included one clinical study [78], and the concurrent treatment group of herbal medicine and acupuncture showed significant pain reduction with an MD of –0.87 (95% CI –1.57, –0.17). The results of the meta-analysis are presented in Appendix A. In both meta-analyses, the small number of studies and the small number of included subjects may raise concerns about imprecision; therefore, the level of evidence was lowered by one notch, and moderate was decided.

Based on the clinical experience of the development committee and the results of studies related to the safety of herbal medicine [74], it was judged that the concurrent treatment of herbal medicine with Korean medicine, such as acupuncture, did not raise safety concerns. Therefore, it was concluded that the clinical benefit was greater than the harm. In addition, a survey of Korean medicine doctors [13] found that herbal medicine is the most representative treatment method for TMDs in clinical practice. Accordingly, the recommended grade was determined as “B”.

### 3.6. Exercise Therapy

Exercise therapy is widely used in clinical practice to treat TMDs. In particular, it is used in movement acupuncture [80,81,82] and Do-in exercise therapy, and because of the characteristics of Korean medicine treatment for TMDs, exercise therapy to correct systemic spine alignment can be performed. Therefore, exercise therapy was not limited to temporomandibular joint stabilization exercise; the effects of various exercise therapies such as Do-in therapy, cervical spine stabilization exercise, and stretching were confirmed, and recommendations were drawn.

#### Exercise Treatment Group vs. Inactive Control Group

Exercise therapy should be considered for symptomatic improvement in patients with TMDs (R12, grade of recommendation: B; level of evidence: moderate).

Five studies [83,84,85,86,87] were selected. The meta-analysis of pain included two studies and showed a significant decrease in MD (–1.99 [95% CI –2.51, –1.46]). Meta-analysis of the extent of mouth opening also included two studies [80,81] and showed a significant increase with an MD of 10.33 (95% CI 1.79, 18.86). A meta-analysis of the improvement rate also included two studies [83,85], and a significant improvement was confirmed in the exercise therapy group, with an RR of 18.14 (95% CI 6.43, 51.23). However, since only two studies [86,87] were included in each meta-analysis, there were concerns about imprecision. Therefore, in all results, the level of evidence was lowered by one notch, and accordingly, all levels of evidence for exercise therapy were determined to be moderate. The meta-analysis results are presented in Appendix A.

In a clinical survey of Korean medicine doctors [13], 19.35% of doctors responded that they were using Do-in therapy. In addition, it was confirmed in the clinical expert interview that clinical practice emphasizes temporomandibular joint stabilization exercises, Do-in therapy with acupuncture, Dong-qi acupuncture, and stretching and exercise for correct spinal alignment. Therefore, it was judged to have high clinical utility. Furthermore, no side effects of exercise were identified in previous reports [83,84,85,86,87], and it was considered that the risk of harm from exercise therapy under the guidance of a Korean medicine doctor was very low. Therefore, the recommended grade was determined as “B”.

### 3.7. Korean Medicine Physical Therapy

In a survey on the clinical status [13] of Korean medicine doctors related to temporomandibular joint disorders, 34.4% answered that they used one or more medications. Interferential current therapy (ICT) is the most commonly used treatment method, followed by transcutaneous electrical nerve stimulation (TENS), ultrasound therapy, and silver spike points (SSPs). Thermotherapy is also widely used.

#### 3.7.1. Korean Medicine Physical Therapy Group vs. Inactive Control Group

In clinical practice for patients with TMDs, physical therapy in Korean medicine should be considered to improve symptoms (R15, grade of recommendation: B; level of evidence: moderate).

Thirty-eight clinical studies related to meridian low-level laser therapy (LLLT) [56,88,89,90,91,92,93,94,95,96,97,98,99,100,101,102,103,104,105,106,107,108,109,110,111,112,113,114,115,116,117,118,119,120,121,122,123,124] and two clinical studies on transcutaneous electrical nerve stimulation (TENS) [120,125] were identified.

##### Meta-Analysis for Meridian Low-Level Laser Therapy (LLLT)

Seventeen clinical studies [89,92,94,96,97,99,102,103,105,109,110,115,119,120,122,124] were included in a pain meta-analysis, and significant improvement was confirmed in the meridian low-level laser therapy (LLLT) group with an SMD of –1.01 (95% CI –1.58, –0.44). Six studies [91,91,94,97,109,121] were included in the meta-analysis of pain changes, and significant improvement was confirmed with an SMD of 0.99 (95% CI 0.69, 1.29). In the meta-analysis of the mouth opening amount, a significant effect was confirmed with an MD of 3.31 (95% CI 2.39, 4.22). In all three meta-analyses, heterogeneity was confirmed by chi-square and I2 tests; therefore, it was judged that the level of evidence needed to be lowered by one level due to concerns about inconsistency. Three studies [91,98,109] were included in the meta-analysis of changes in mouth opening, and a significant increase was confirmed. Although there was no heterogeneity, the level of evidence was lowered by one level because three studies were included, and there were concerns about imprecision due to the small number of study subjects. Therefore, the level of evidence for pain, pain change, mouth opening, and change in mouth opening after meridian low-level laser therapy (LLLT) was determined to be moderate. The meta-analysis results are presented in Appendix A.

##### Meta-Analysis for Transcutaneous Electrical Nerve Stimulation (TENS)

A meta-analysis of pain included two studies [120,125], and significant pain reduction was confirmed. However, considerable heterogeneity was observed, and there were concerns about inconsistency and imprecision because the two studies were conducted with small numbers of subjects. Therefore, the level of evidence was downgraded to two notches, and the level of evidence was determined to be low. In addition, only one study [120] was included in the meta-analysis of mouth opening. The meta-analysis results showed a significant increase in the MD (6.15 [95% CI 4.42, 7.88]). However, because only one study was included, there were concerns about imprecision; therefore, the level of evidence was moderate. The meta-analysis results are presented in Appendix A.

##### Recommendation

Both meridian low-level laser therapy (LLLT) and transcutaneous electrical nerve stimulation (TENS) have clinical benefits for pain management and mouth opening. After collecting opinions on the clinical status and clinical experience of experts, it was determined that there were no major concerns about safety. Therefore, we conclude that the benefits outweigh the harm caused. The use of Korean physical therapy has been reported to be very high in clinical practice, and transcutaneous electrical nerve stimulation (TENS) has been actively used in clinical practice. Meridian low-level laser therapy (LLLT) has not yet been actively used in Korean medical institutions; therefore, its clinical use of meridian low-level laser therapy (LLLT) was judged to be moderate. Therefore, the development committee decided on the recommended grade being B.

#### 3.7.2. Concurrent Treatment Group of Korean Physical Therapy and Usual Conservative Treatment vs. the Usual Conservative Treatment Group

In the clinical treatment of TMJ patients with temporomandibular joints receiving the usual conservative treatment, Korean medical physical therapy should be considered to improve symptoms (R16, grade of recommendation: B; level of evidence: moderate). 

Two studies [111,126] identified concurrent treatment with transcutaneous electrical nerve stimulation (TENS) and conventional conservative treatment, one study [111] identified concurrent treatment with meridian low-level laser therapy (LLLT) and usual conservative treatment, and one study [127] identified concurrent treatment with ultrasound and usual conservative treatment. Meta-analysis results on pain reduction in two studies [126,128] were included, with an MD of 0.64 (95% CI 0.13, 1.16), and the group treated with Korean herbal physiotherapy showed a significant pain reduction (Appendix A). However, there were two selected references [126,128], and the level of evidence was determined to be moderate because the number of study subjects was small, and there were concerns about imprecision.

Although it was difficult to judge the risk because side effects were not reported in the included studies, through the clinical experience of the development committee, it was concluded that concurrent treatment with Korean physical therapy, such as electrical stimulation, with usual conservative treatment, does not raise safety concerns. Therefore, the development committee judged that the clinical benefit was higher than the harm and assigned the grade of recommendation of “B”.

### 3.8. Korean Medicine Combination Treatment

#### Korean Medicine Combination Treatment Group Consisting of Acupuncture, Herbal Medicine, Chuna, etc., and the Usual Conservative Treatment Group

In the clinical treatment of patients with TMDs, the use of Korean medicine combination treatment, rather than the usual conservative treatment, should be considered to improve symptoms (R17, grade of recommendation: B; level of evidence: moderate). 

One document [129] was identified as randomized controlled trials (RCTs) conducted in the United States. In total, 92 patients were included, and the control group was treated with the usual conservative treatment by a dentist specializing in TMD management. Acupoint and herbal medicines were selected according to the pattern identification for Korean medicine combination treatment. A meta-analysis was performed on the VAS scores for severe and average pain. Significant improvement was confirmed in the Korean medicine combination treatment group with meta-analysis results in MD (−1.20 [95% CI −2.19, −0.21]) for severe pain measured with VAS. In addition, a significant improvement was confirmed in the Korean medicine combination treatment group, with meta-analysis results with an MD of −1.00 (95% CI −1.84, −0.16) for mean pain measured with VAS. However, because only one study was included, there were concerns regarding imprecision. Therefore, the level of evidence was lowered by one notch and accordingly determined to be moderate. The meta-analysis results are presented in Appendix A.

When checking the safety reports in the included studies, there were no severe side effects in either group, and only minor side effects were confirmed, which were reported to have disappeared within a short period. Therefore, it was judged that the benefit was more significant than the harm caused by the combination treatment with Korean medicine. Therefore, the recommended grade was determined as “B”.

### 3.9. Intraoral Balancing Appliance

The temporomandibular joint intraoral balancing appliance is an established treatment method rooted in classical Korean medicine [130]. It is usually applied as a mouthguard between the upper and lower teeth to provide balance between the temporomandibular joints. This method is actively employed to treat TMDs, as 37.63% of Korean medicine doctors [10] have experience in treating temporomandibular joints using the intraoral balancing appliance. A case report served as the foundation for a study that utilized this appliance for TMD treatment. Furthermore, noteworthy treatment outcomes were documented in studies [131,132,133,134]. However, the absence of randomized controlled trials (RCTs) prevents the provision of corroborative references.

Considering the clinical experience and case reports to date, the clinical benefits are expected to be significant. Furthermore, considering that the intraoral balancing appliance is a flexible device, it was judged through an interview with a clinical expert that the possibility of harm caused by this device would be very low if it were worn only for a specific time a day within three months. Therefore, based on the clinical experience of the development committee, clinical considerations were derived for the implementation of an intraoral balancing appliance for adult TMDs.

### 3.10. Thread-Embedding Acupuncture

No clinical studies satisfying the reference selection or study design criteria were identified. However, based on survey results [13] and interviews with clinical experts, thread-embedding acupuncture treatment has recently expanded its scope of use in Korean medicine, and its clinical use is also increasing.

Thread-embedding acupuncture treatment requires sufficient education and skills during the procedure. For safe thread-embedding acupuncture, it may be necessary to establish and follow guidelines for sterilization, disinfection, and aseptic methods [135]. Under the condition that guidelines and education on sterilization, disinfection, and aseptic procedures are well established, the clinical benefit of thread-embedding acupuncture treatment could outweigh the harm. Therefore, thread-embedding acupuncture can be considered for adult patients with TMDs.

The finally derived recommendations are presented in Table 3.

## 4. Discussion

In Korea, the prevalence of chronic TMD for more than three months is 3.1%, making it a common disease [136]. TMDs are defined as “musculoskeletal and neuromuscular symptoms in the temporomandibular joint, masticatory muscles, and related tissues” [137]. Recent studies tend to view TMDs as complex disorders of the masticatory organs [138]. The most common symptoms of TMDs are pain in the ears, head, and face, including limited range of motion and clicking sounds [139]. Additionally, TMDs have a significant impact on quality of life [140]. The diagnosis of TMDs is made through the evaluation of medical history and physical examination, and a representative diagnostic tool is the Diagnostic Criteria for TMDs (DC/TMD) developed in 2014 [141].

The treatment of patients with TMDs aims to improve functions such as increasing the range of motion of the temporomandibular joint and relieving pain and inflammation in the temporomandibular joint and masticator muscles [142]. Therefore, in the KMCPG for the temporomandibular joint, we analyzed the effects of interventions on pain reduction, functional improvement, and quality of life improvement to prepare recommendations. If possible, the effects on pain, functional improvement, and quality of life improvement during the meta-analysis were analyzed by dividing the patients into subgroups.

Acupuncture is a treatment method in which needles are inserted into the skin and underlying tissues of a specific area known as an acupuncture point (穴位) or spots on the body suitable for acupuncture (經穴) [143,144]. Acupuncture is the most common medical treatment for musculoskeletal disorders in Korea. It controls pain by regulating endogenous opioids, serotonin, and norepinephrine mainly through nociceptors, inflammatory cytokines, and central nervous system activation [145,146]. We found that the number of acupuncture research references on temporomandibular joint disease included in this study and the number of study subjects included were sufficient, the level of evidence of the studies was high, and the effect of acupuncture was significantly higher than that in the control group. Therefore, the grade of recommendation for acupuncture in TMDs was determined to be “A” (highest level).

Acupuncture for TMDs is performed by choosing between distal and local acupuncture. In clinical studies on acupuncture related to TMDs [21,22,23,24,25,26,27,28,29], acupoints around the temporomandibular joint are often used. However, in a Korean medicine doctor survey on the clinical status of TMDs [13], there were Korean medicine doctors who responded that meridian pattern identification was significant during treatment. Considering this, it is presumed that many distal acupoints were selected. Therefore, to address these clinical questions, related data were reviewed. The results confirmed that distal acupuncture, proximal acupuncture, and distal and proximal combined acupuncture had similar pain and functional improvement effects on TMDs in adults.

In clinical situations, it is often necessary to determine whether additional Chuna manipulation and herbal medicine treatments should be administered to patients undergoing acupuncture. In addition, many patients visit Korean medical institutions while undergoing the usual conservative treatment at Western hospitals. Recommendations have been made to address these clinical questions. Concurrent treatment with Chuna manipulation or herbal medicine treatment with Korean medicine, such as acupuncture, resulted in significant pain and functional improvement compared with non-concurrent treatment. When Korean medical treatments, such as acupuncture, are performed for patients with TMDs, additional therapy with Chuna manipulation or herbal medicine should be considered. In patients undergoing conventional conservative treatment, concurrent treatment with acupuncture, Chuna manipulation, herbal medicine, or Korean medicine physical therapy showed significant improvement in pain and function compared to the usual conservative treatment alone. Therefore, Korean medicine should be considered for patients with TMDs who receive only conservative treatment.

When a patient with chronic or severe TMD visits a Korean medical institution, acupuncture, Chuna manipulation, and herbal medicine are applied in combination with Korean medicine. Recommendations have been made to reflect these clinical realities. It was concluded that combination treatment with Korean medicine should be considered rather than the usual conservative treatment for symptom improvement in the clinical treatment of patients with TMDs.

In the case of pharmacopuncture and Chuna manipulation in the pre-certified KMCPG [3] development study of the temporomandibular joint, there was a limitation in deriving recommendations owing to the lack of clinical studies. In addition, clinical research on pharmacopuncture and herbal medicine treatment has not been actively conducted in Korea owing to institutional limitations. Therefore, to draw recommendations based on high-quality clinical research results, we conducted a clinical study to provide evidence of the efficacy of pharmacopuncture and Chuna manipulation. In the case of pharmacopuncture, multicenter randomized controlled trials (RCTs) [5] have been conducted, in which Zahageo pharmacopuncture was set as the treatment group after the Investigational New Drug (IND) approval of the Ministry of Food and Drug Safety (MFDS). Chuna manipulation was also conducted in a multicenter randomized clinical trial to evaluate the treatment effects, safety, and economic feasibility. By reflecting on these results, recommendations with a high level of evidence can be drawn [4].

Herbal medicines are one of the most important treatments for TMDs. However, there were difficulties in securing evidence because clinical studies were not conducted due to institutional limitations. Nevertheless, if clinical research related to herbal medicines becomes more active in the future, this limitation can be overcome.

Intraoral balancing appliances and thread-embedding acupuncture are actively used in Korean clinical practice. However, it was impossible to make a recommendation because there were no studies that could be used as a basis for drawing guidance. According to the link between the grade of recommendation and the level of evidence, the grade of recommendation GPP was considered through expert consensus. However, according to the current development methodology, a classical literature basis should be recorded for herbal medicines prescribed by the Ministry of Health and Welfare (MOHW) and the Ministry of Food and Drug Safety (MFDS) notification. However, these classical reference data could not be obtained because of the interventional characteristics of the intraoral balancing appliance and thread-embedding acupuncture. Therefore, it is not easy to draw a recommendation for the grade of GPP. As a result, the development committee decided to recommend intraoral balancing appliance and thread-embedding acupuncture treatment based on clinical considerations considering the clinical application. Through future clinical studies, it is necessary to establish evidence for the effects of intraoral balancing appliances and thread-embedding acupuncture treatment for TMDs, and systematic and continuous research on safety and side effects is required in the future.

This recommendation has some limitations. First, the level of evidence had to be lowered in many cases due to the risk of bias, inconsistency, and inaccuracy. Second, owing to the small number of published studies and the lack of evidence to support their use, some clinical questions had to be answered through expert consensus. Third, placebo effects cannot be completely ruled out in some interventions because of the difficulty of blinding the interventions. Future RCTs with placebo controls will solve this problem.

This is the first KMCPG for temporomandibular joint disorders using an evidence-based research methodology, such as a systematic reference review. By revising this KMCPG, it is intended to help Korean medicine doctors and patients in the treatment of TMDs in decision making and contribute to improving public health and strengthening the insurance coverage of Korean medicine through the implementation of standardized Korean medicine treatment.

## 5. Conclusions

As a result of developing the KMCPG for TMDs through an evidence-based research methodology, the following conclusions were drawn:

Acupuncture, pharmacopuncture, and Chuna manual therapy are recommended for TMD patients in clinical practice. Concurrent conventional conservative therapy with Korean medicine or a combination of Korean medicines should be considered in clinical practice in patients with temporomandibular disorders. This updated guideline will provide Korean medicine clinicians with evidence-based information when treating patients.

## Figures and Tables

**Table 1 healthcare-11-02364-t001:** Level of evidence.

Level of Evidence	Meaning
High	It can be very confident that the estimate of the effect is close to the actual effect.
Moderate	Confidence in the estimate of the effect can be moderate. Estimates of the effect are likely to be close to the actual effect but may differ significantly.
Low	Confidence in the estimate of the effect is limited. Actual effects may differ significantly from estimated effects.
Very Low	There is little confidence in the estimate of the effect. The actual effect will be quite different from the estimate of the effect.

**Table 2 healthcare-11-02364-t002:** Grade of recommendation.

Grade of Recommendation	Meaning	Notation
A	It is recommended when the benefits are obvious, and the clinical use is high.	Is recommended
B	It is granted when the benefits are reliable and the use is high or moderate in the medical field or when clinical benefits are evident, even if the evidence for the study related to the recommendation is insufficient.	Should be considered
C	It is granted when the benefit is unreliable, but the utilization is high or moderate in the medical field.	May be considered
D	The benefits are unreliable and can lead to harmful consequences.	Is not recommended
GPP	Based on the bibliographic basis, recommendations are made based on the official agreement of the expert group.	Is recommended based on the expert group consensus

GPP: Good Practice Point.

**Table 3 healthcare-11-02364-t003:** Grade of recommendation/level of evidence of Korean medicine for temporomandibular disorders.

Rec No	Description of Rec	Grade of Rec./Level of Evidence
**(1) Acupuncture**		
R1	Acupuncture treatment is recommended for clinical practice of TMD patients.	A/High
R1-1	Remote or neighboring acupuncture points should be considered according to the judgment of KMDs in consideration of the clinical condition of the patient in the clinical practice of TMD patients.	B/Moderate
R2	Acupuncture treatment should be considered in clinical practice for TMD patients.	B/Moderate
R3	Concurrent treatment with acupuncture should be considered in clinical practice for symptom improvement in TMD patients undergoing the usual conservative treatment.	B/Moderate
**(2) Laser acupuncture**		
R4	Laser acupuncture treatment may be considered in the clinical practice of TMD patients for the improvement of symptoms.	C/Low
**(3) Pharmacopuncture**		
R5	Consideration of pharmacopuncture treatment is recommended in the clinical practice of TMD patients for the improvement of symptoms.	A/Moderate
**(4) Chuna manual therapy**		
R6	Chuna manual therapy is recommended in the clinical practice of TMD patients for the improvement of symptoms.	A/High
R7	Concurrent treatment with Chuna manual therapy should be considered in clinical practice for symptom improvement in TMD patients undergoing the usual conservative treatment.	B/Moderate
R8	Concurrent treatment with Chuna manual therapy should be considered in clinical practice for symptom improvement in TMD patients undergoing Korean medicine treatment including acupuncture.	B/Moderate
**(5) Herbal medicine**		
R10	Concurrent treatment with herbal medicine treatment should be considered in clinical practice for symptom improvement in TMD patients undergoing the usual conservative treatment.	B/Moderate
R11	Concurrent treatment with herbal medicine treatment should be considered in clinical practice for symptom improvement in TMD patients undergoing Korean medicine treatment including acupuncture.	B/Moderate
**(6) Exercise therapy**		
R12	Exercise therapy should be considered in the clinical practice of TMD patients for the improvement of symptoms.	B/Moderate
**(7) Korean medicine physiotherapy**		
R15	Korean medicine physiotherapy should be considered in the clinical practice of TMD patients for the improvement of symptoms.	B/Moderate
R16	Concurrent treatment with Korean medicine physiotherapy should be considered in the clinical practice for symptom improvement in TMD patients undergoing the usual conservative treatment.	B/Moderate
**(8) Combination of Korean medicine treatments**		
R17	A combination of Korean medicine treatments should be considered rather than the usual conservative treatment in the clinical practice of TMD patients for the improvement of symptoms.	B/Moderate

Rec.: recommendation; No.: number; TMD: temporomandibular disorder.

## Data Availability

All relevant data have been included in the manuscript and Appendix A.

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
