# Peer review of "Korean Medicine Clinical Practice Guideline Update for Temporomandibular Disorders: An Evidence-Based Approach"

_healthcare, 2023, doi:10.3390/healthcare11162364_

Round 1
Reviewer 1 Report
This fascinating paper needs to have the text improved so that it can be more easily appreciated outside of Korea, if it is to be published in a journal with a world-wide readership. At present it mainly makes sense to Korean readers.
1. Is Korean Medicine a special field or just medicine in Korea ? Explain please .
2. Explain "oral balance device " Is this a mouth guard or occlusal cover ?
3. Explain Chuna Therapy please ? Most readers have never heard of it.
4. Explain "usual conservative treatment" in 3.1.3
5. Lines 64-67 - what currency ? and give US $ equivalents.
6. I think that there is a lot of repetition in the Methods and Results sections, these could be shortened. Indeed, Table 3 appears to summarise all the results ; and would be better placed at the end of the Results, rather than at the end of the Methods section.
7. The Discussion does not discuss the role of possible placebo effects for the various complimentary therapies.
8. Many of the references quoted are in journals which the average reader will not easily access. That does not rule them out, but it is worth some discussion. Also what contrasts are there with treatments commonly used in (say) Europe and the Americas ?
9. Please complete the Acknowledgements and Conflicts of Interests sections.
In summary, this reviewer would shorten the text in the Results section by perhaps 10-25%, perhaps by greater use of Tables, and make sure that some of these treatments are explained in a manner that all readers of the Journal can understand.
Author Response
We thank the reviewers for their thorough assessment and thoughtful suggestions, which helped improve this manuscript. We have incorporated the suggestions in the revised version of this manuscript, and a point-by-point response to the reviewer’s comments is presented below. After checking the revision messages, our manuscript went through additional language editing by native speaker because one of the reviewers pointed out minor language issues (certificated attached).
- Is Korean Medicine a special field or just medicine in Korea ? Explain please .
- Korean Medicine is a special field which has been traditionally conducted. We added explanation in the head of Introduction section.
- Explain "oral balance device " Is this a mouth guard or occlusal cover ?
- Oral balance device is a kind of mouth guards. It is kept between upper teeth and lower teeth to give balance between both temporomandibular joints. We added description in 3.9 Intraoral Balancing Appliance section.
- Explain Chuna Therapy please ? Most readers have never heard of it.
- It is a manual therapy that is originated from Korean Medicine and it includes manipulation, myofascial release, joint mobilization, and so on. We added explanation about Chuna Therapy in 3.4 Chuna Manipulation section.
- Explain "usual conservative treatment" in 3.1.3
- The examples of ‘usual conservative treatment’ has been in the Materials and Methods section, 2.6. Selection of Key Questions.
- Lines 64-67 - what currency ? and give US $ equivalents.
- It is Korean currency, ‘won.’ To make it widely understood as you pointed out, we added US dollar corresponding to Korean won.
- I think that there is a lot of repetition in the Methods and Results sections, these could be shortened. Indeed, Table 3 appears to summarise all the results ; and would be better placed at the end of the Results, rather than at the end of the Methods section.
- We erased repeated contents from the Methods and Results section. We moved Table 3 to the end of the Results section.
- The Discussion does not discuss the role of possible placebo effects for the various complimentary therapies.
- literatures regarding acupuncture had some placebo group as control but other interventions like Chuna, intraoral balance device, etc. did not have placebo control. So, we added this explanation in the limitation paragraph in the Discussion section.
- Many of the references quoted are in journals which the average reader will not easily access. That does not rule them out, but it is worth some discussion. Also what contrasts are there with treatments commonly used in (say) Europe and the Americas ?
- There were many Chinese literatures cited as you pointed out. Interventions of Korean Medicine are acupuncture, pharmacopuncture, Chuna, etc. and these are considerably conducted in East Asian country. The way these are performed is similar to those in Europe and America because they are originated from East Asian countries.
- Please complete the Acknowledgements and Conflicts of Interests sections.
- Thank you for giving us notice. We completed the Acknowledgements and Conflicts of Interests sections.
In summary, this reviewer would shorten the text in the Results section by perhaps 10-25%, perhaps by greater use of Tables, and make sure that some of these treatments are explained in a manner that all readers of the Journal can understand.
- We agreed with what you pointed out. We shortened the Methods and Results section and added some explanations to be easily understood by readers all over the world. If you think it is not appropriately modified, please let us know.
Reviewer 2 Report
Suggested: Accepted with Minor revision
This review with relative update clinical guideline of Korean medicine for treating TMDs is very interesting and methodologically well written. However, there are some issues to be responded, especially the lack of all the data and papers reviewed (appendix) and the lack of several references reported and numbered in the text. The major points to address are as follows:
- TITLE: “teporomandibular” is written in the wrong way, please correct
- 2.7. Reference Search and Selection (Line 179): why did the authors not consider the search in Pubmed and Scopus among Western medical literature DBs? please justify
- Line 181-182: please specify the names of all Chinese and Japanese DBs used for the literature search, as for the others previous mentioned
- Table 1-2-3 (Line 231-234/249-251): the graphics of the tables are not clear, resulting difficult to read. Please edit (for example by inserting a space between rows so that they are easier to understand)
- Line 237-238: “The consensus was reached on the recommendation, grade of recommendation, and level of evidence through the Delphi technique, an official consensus method.” – Please provide evidence
- Table 3 (Line 247-251): this table shows the final grade of recommendation, so it needs to be moved to the results section
- Line 392: please provide also the names of the acupoints (Imun (耳門), Hagwan (下關), and Yepung (翳關) according to the standard international acupuncture nomenclature
- Supplementary materials (Line 827): supplementary materials are missing. Please provide the appendix 1-17 mentioned in the text, with all results exposed and the characteristics of the paper analysed in this revision. According to this, a table with a summary of all the papers considered and the relative kind of study, treatment used etc. is strongly recommended
- Acknowledgments and Conflicts of Interest (line 830-850): please fill in with the appropriate statements
- References: the references from 112 onward are missing. Please provide
Author Response
We thank the reviewers for their thorough assessment and thoughtful suggestions, which helped improve this manuscript. We have incorporated the suggestions in the revised version of this manuscript, and a point-by-point response to the reviewer’s comments is presented below. After checking the revision messages, our manuscript went through additional language editing by native speaker because one of the reviewers pointed out minor language issues (certificated attached).
Suggested: Accepted with Minor revision
This review with relative update clinical guideline of Korean medicine for treating TMDs is very interesting and methodologically well written. However, there are some issues to be responded, especially the lack of all the data and papers reviewed (appendix) and the lack of several references reported and numbered in the text. The major points to address are as follows:
- TITLE: “teporomandibular” is written in the wrong way, please correct
- We changed it according to the right spelling. Thank you very much.
- 2.7. Reference Search and Selection (Line 179): why did the authors not consider the search in Pubmed and Scopus among Western medical literature DBs? please justify
- Cochrane handbook stated that “The three bibliographic databases generally considered to be the most important sources to search for reports of trials are CENTRAL (Cochrane library) MEDLINE and Embase.” In the Cochrane Handbook for systematic literature review, the scopus DB is not listed as a search database. We searched the most essential DB that was suggested(https://training.cochrane.org/handbook/current/chapter-04#section-4-3).
Cochrane handbook also stated that “Ovid MEDLINE covers all of the available content and metadata in PubMed with a delay of one working day”. Therefore, the Ovid-Medline that we searched is the same source of information from PubMed.
- Line 181-182: please specify the names of all Chinese and Japanese DBs used for the literature search, as for the others previous mentioned
- We searched for China National Knowledge Infrastructure (CNKI) of Chinese database and CiNii of Japanese database. We added the information in the Methods section on the manuscript
- Table 1-2-3 (Line 231-234/249-251): the graphics of the tables are not clear, resulting difficult to read. Please edit (for example by inserting a space between rows so that they are easier to understand)
- According to your opinion, we modified it to be easily understood.
- Line 237-238: “The consensus was reached on the recommendation, grade of recommendation, and level of evidence through the Delphi technique, an official consensus method.” – Please provide evidence
- We attached a file for the consensus process (The file is not for publication). Please check this and if there is not enough information, let us know. We are very enthusiastic to provide any information.
- Table 3 (Line 247-251): this table shows the final grade of recommendation, so it needs to be moved to the results section
- According to your recommendation, we moved the table to the Results section.
- Line 392: please provide also the names of the acupoints (Imun (耳門), Hagwan (下關), and Yepung (翳關) according to the standard international acupuncture nomenclature
- We inserted internationally standard names of acupoint consisting of alphabet and number.
- Supplementary materials (Line 827): supplementary materials are missing. Please provide the appendix 1-17 mentioned in the text, with all results exposed and the characteristics of the paper analysed in this revision. According to this, a table with a summary of all the papers considered and the relative kind of study, treatment used etc. is strongly recommended
- We added the information in supplementary materials according to your comment.
- Acknowledgments and Conflicts of Interest (line 830-850): please fill in with the appropriate statements
- We filled Acknowledgments and Conflicts of Interest with appropriate statements.
- References: the references from 112 onward are missing. Please provide
- There were errors regarding the references. We changed it correctly.
Reviewer 3 Report
Dear Authors,
The topic of this article may be scientifically and clinically interesting, helping many specialists to diagnose and to treat TMDs. Only a few suggestions may be considered as follows:
Title
The title should be more precise and informative. I suggest to specify that it is a systematic review and meta-analysis.
Keywords
The keyword “therapeutic options” or “conservative therapy” should be added.
Introduction
The introduction is well developed.
Please, use the acronyms “TMD” and “CPG” in the whole text, as in the abstract.
Some references should be added in this section, especially regarding the drug therapy or behavioral therapy.
For example, this reference could be added in line 68:
- Montinaro F, Nucci L, d'Apuzzo F, Perillo L, Chiarenza MC, Grassia V. Oral nonsteroidal anti-inflammatory drugs as treatment of joint and muscle pain in temporomandibular disorders: A systematic review. Cranio. 2022 Feb 7:1-10. doi: 10.1080/08869634.2022.2031688.
Materials and methods
Tables 1 - 2 and 3 need to be revised; they are a lit messy. Please reorder the tables to offer better reading to readers.
Discussions
Discussion section is well written. However, there are sometimes repetitions of the results and this may be reviewed.
Conclusions
Further discussion should be done in the conclusions to better insert the research in the right perspective.
Dear Authors,
I suggest a minor editing of English language.
Author Response
We thank the reviewers for their thorough assessment and thoughtful suggestions, which helped improve this manuscript. We have incorporated the suggestions in the revised version of this manuscript, and a point-by-point response to the reviewer’s comments is presented below. After checking the revision messages, our manuscript went through additional language editing by native speaker because one of the reviewers pointed out minor language issues (certificated attached).
Dear Authors,
The topic of this article may be scientifically and clinically interesting, helping many specialists to diagnose and to treat TMDs. Only a few suggestions may be considered as follows:
Title
The title should be more precise and informative. I suggest to specify that it is a systematic review and meta-analysis.
- Thank you for your opinion. This guideline is suggested based on systematic review and meta-analysis, but this is not systematic review and meta-analysis itself. Therefore, we had put the phrase like ‘clinical practice guideline’ in the title. If I did not understand what you pointed out, please let me know. We authors are eager to modify it according to your saying.
Keywords
The keyword “therapeutic options” or “conservative therapy” should be added.
- We added those two keywords according to your opinion. Thank you very much.
Introduction
The introduction is well developed.
Please, use the acronyms “TMD” and “CPG” in the whole text, as in the abstract.
- We changed those two words with acronyms according to your opinion.
Some references should be added in this section, especially regarding the drug therapy or behavioral therapy.
For example, this reference could be added in line 68:
- Montinaro F, Nucci L, d'Apuzzo F, Perillo L, Chiarenza MC, Grassia V. Oral nonsteroidal anti-inflammatory drugs as treatment of joint and muscle pain in temporomandibular disorders: A systematic review. Cranio. 2022 Feb 7:1-10. doi: 10.1080/08869634.2022.2031688.
- We added some references including the literature you took as an example according to your opinion
Materials and methods
Tables 1 - 2 and 3 need to be revised; they are a lit messy. Please reorder the tables to offer better reading to readers.
- We revised those tables to be better understood by readers.
Discussions
Discussion section is well written. However, there are sometimes repetitions of the results and this may be reviewed.
- We shortened Discussion section by erasing some repeated contents.
Conclusions
Further discussion should be done in the conclusions to better insert the research in the right perspective.
- We authors discussed the meaning of this clinical practice guideline and modified the conclusion to better insert this research in the right perspective. If you think it is not appropriately modified, please let us know.
Dear Authors,
I suggest a minor editing of English language.
- This manuscript had been edited by native English speaker. However, we tried the additional editing service again to make it better according to a reviewer’s comment regarding minor language issue. We attached the certificate for English editing.
Reviewer 4 Report
The manuscript summarized the current knowledge regarding conservative treatment of TMJ, including Korean medicine or acupunture. The manuscript is well-documented, refernces are adequate, however relativly few publication were cited from the last decade. I recommend the manuscript for publication.
Author Response
We thank the reviewers for their thorough assessment and thoughtful suggestions, which helped improve this manuscript. We have incorporated the suggestions in the revised version of this manuscript, and a point-by-point response to the reviewer’s comments is presented below. After checking the revision messages, our manuscript went through additional language editing by native speaker because one of the reviewers pointed out minor language issues (certificated attached).
The manuscript summarized the current knowledge regarding conservative treatment of TMJ, including Korean medicine or acupunture. The manuscript is well-documented, refernces are adequate, however relativly few publication were cited from the last decade. I recommend the manuscript for publication.
- Thank you very much. We added some recent references according to your opinion.
Round 2
Reviewer 1 Report
I thank the authors for this revision. The paper is much easier to read now and I am happy to recommend that it is accepted.